# Tuning flavin environment to detect and control light-induced conformational switching in Drosophila cryptochrome

Siddarth Chandrasekaran [1], Connor M. Schneps[1], Robert Dunleavy[1], Changfan Lin[1], Cristina C. DeOliveira[1], Abir Ganguly [2] & Brian R. Crane [1✉]

Light-induction of an anionic semiquinone (SQ) flavin radical in Drosophila cryptochrome (dCRY) alters the dCRY conformation to promote binding and degradation of the circadian clock protein Timeless (TIM). Specific peptide ligation with sortase A attaches a nitroxide spin-probe to the dCRY C-terminal tail (CTT) while avoiding deleterious side reactions. Pulse dipolar electron-spin resonance spectroscopy from the CTT nitroxide to the SQ shows that flavin photoreduction shifts the CTT ~1 nm and increases its motion, without causing full displacement from the protein. dCRY engineered to form the neutral SQ serves as a dark-state proxy to reveal that the CTT remains docked when the flavin ring is reduced but uncharged. Substitutions of flavin-proximal His378 promote CTT undocking in the dark or diminish undocking in the light, consistent with molecular dynamics simulations and TIM degradation activity. The His378 variants inform on recognition motifs for dCRY cellular turnover and strategies for developing optogenetic tools.

[1] Department of Chemistry and Chemical Biology, Cornell University, Ithaca, NY, USA. [2] Institute for Quantitative Biomedicine, Rutgers University, Piscataway, NJ, USA. ✉email: bc69@cornell.edu

Cryptochromes (CRYs) serve as blue-light sensors in species that range from bacteria to vertebrates[1,2]. CRYs play important roles in a number of sensory and regulatory functions including inhibition of hypocotyl elongation in plants[3], blue-light evoked depolarization of neurons[4], and the oscillation and entrainment of circadian clocks[1,2,4]. CRYs belong to the same protein superfamily as photolyases, a class of enzymes that repair ultraviolet (UV)-damaged DNA in a light-dependent fashion[5]. Structurally, all CRYs contain both a conserved photolyase homology region (PHR) with the capacity to bind flavin adenine dinucleotide (FAD) and a CRY C-terminal extension (CCE) of variable length and sequence that often regulates interactions with other proteins[6–10]. In type I insect CRYs the 23-residue CCE ends in a highly conserved 10-residue helical C-terminal tail (CTT) that inserts into the flavin binding pocket[6,7]. The signaling properties of CRYs depend strongly on the CCE conformation; however, structural changes in the tail domain have been difficult to monitor directly by biophysical techniques owing to its dynamic nature in activated states.

In the fly Drosophila melanogaster, cryptochrome (dCRY) entrains the circadian clock to light[2,7–11]. The Drosophila clock is comprised of a negative feedback loop in which the core oscillator proteins Timeless (TIM) and Period (PER) form a heterodimeric complex that represses expression of their respective genes by binding to a transcriptional activator complex composed of Clock and Cycle[1]. Blue light photoreduces the oxidized flavin cofactor of dCRY to the anionic semiquinone (ASQ) state[12–15], which induces a conformational change in the CTT that promotes binding to TIM[15–18]. Light activation also recruits the E3 ligase Jetlag (JET)[18,19], which leads to the degradation of TIM. After light activation, dCRY itself undergoes targeted degradation involving JET[18,19] and also the E3 ligase BRWD3[20]. Thus, by responding to environmental light, dCRY controls the level of cytoplasmic PER:TIM through its light-dependent CTT release and regulates its own turnover. Although mammalian CRY1 proteins do not sense light, regions in their CCEs similarly mediate interactions between the PHRs and other signaling partners[21]. The long CCEs of plant CRYs also respond to light[22,23].

Conformational change in the dCRY CTT correlates with photoreduction of the flavin to the ASQ[12–15]. Alterations to a conserved tetrad of tryptophan residues that supply electrons to the photoexcited flavin[24,25] affect dCRY photoreduction rates, dCRY light stability, and TIM degradation activity[26]. However, some studies have found that Phe substitutions in the Trp tetrad that do not photoreduce retain activity in several assays[12,27–29]. Thus, a better understanding of the dCRY photocycle requires connecting functionally relevant conformations to specific flavin states. The dark-state structure of dCRY shows that the CTT binds to the FAD pocket through a conserved FFW motif (F534, F535, and W536), which contributes to dCRY stability[6,7,15,30]. A conserved histidine residue, H378, separates the FAD cofactor and the FFW motif (Fig. 1). Molecular dynamics (MD) simulations suggest that the negative charge on the ASQ promotes H378 protonation, which in turn disrupts interactions between the FFW motif and the flavin pocket[13]. Residue substitutions to H378 affect TIM degradation in cell culture[13], confirming that H378 is a critical residue for regulating CTT release. Interestingly, mutations of H378 to R, or K, prevent light-induced CRY degradation[13]. However, the H378A dCRY variant, which cannot undergo protonation of residue 378, shows light-induced changes in time-resolved small-angle X-ray scattering profiles very similar to those of wild-type (WT)[31]. Overall, direct monitoring of the CTT with respect to flavin chemical state would clarify the mechanism by which dCRY is conformationally activated. Control of the CTT response would be advantageous for the development of optogenetic tools.

We have developed a spin-labeling strategy to follow CTT dynamics with pulsed electron spin resonance (ESR) spectroscopy. Variant dCRY proteins designed to form the neutral SQ (NSQ) instead of the ASQ do not release the CTT and provide a proxy for the dCRY dark state. Our results give direct observation of CTT release and show that release depends on ASQ formation. H378 proves to be a critical residue for coupling the flavin redox state to CTT conformation and hence signal transduction.

## Results

**An NSQ forming dCRY variant as a dark state proxy.** Probing the dCRY CTT conformation with pulse dipolar ESR spectroscopy (PDS) requires two stable unpaired electron spins separated by ~1–8 nm[32,33]. A typical nitroxide moiety targeted to the dCRY CTT can serve as the first spin, whereas, in principle, the flavin SQ can serve as the second. In dCRY, the ASQ readily forms by

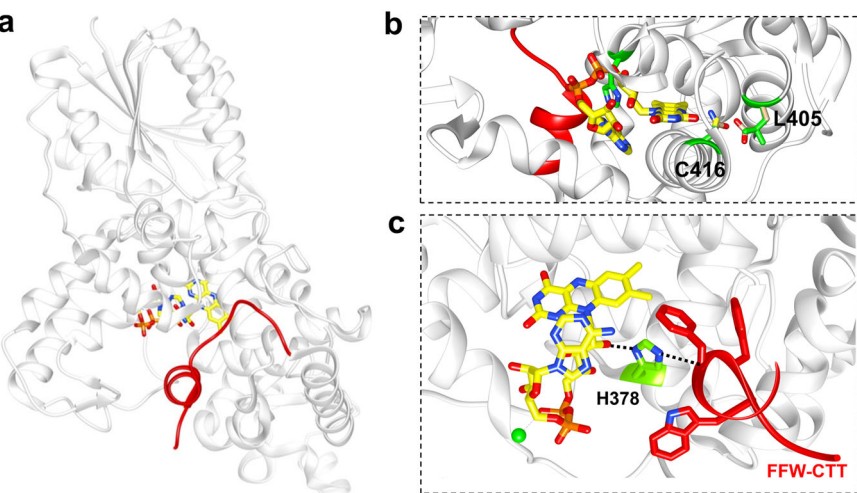

**Fig. 1 Structure of dCRY. a** The C-terminal tail (CTT) of dCRY (red) binds into the pocket containing FAD (yellow). **b** C416 (green) interacts with the FAD N[5] to disfavor protonation, whereas L405 blocks the solvent from accessing the flavin. The N416 and E405 substitutions (tan) favor NSQ formation. The L405E/C416N replacements are located at the distal end from the CTT with the FAD group sandwiched in-between the two sites. **c** H378 (green) bridges the flavin (magnesium shown as green spheres) to the FFW motif of the CTT.

photoreduction and thus is a convenient spin probe for the "light" state[15,34]. However, the diamagnetic "dark" state flavin (FAD[ox]) is not ESR active. Our previous work suggests that negative charge is a key feature of the ASQ for displacing the CTT[13]. We thus reasoned that dCRY containing the NSQ would still bind the CTT and thereby act as a functional mimic of the dark state. We sought to engineer an NSQ-forming dCRY based on the properties of other CRY proteins that preferentially form the NSQ[34–36]. Plant orthologs of dCRY, such as the *Arabidopsis thaliana* cryptochrome 1 (AtCRY1) or *Chlamydomonas reinhardtii* animal-like cryptochrome (aCRY), as well as the ClCRY4 protein from *Columba livia*, either photoreduce to the NSQ or naturally bind the NSQ in the dark[34–36]. Although the overall structures of these orthologs are similar to dCRY, the FAD-binding environments differ in small but significant ways. The SQ protonation state depends on residues that interact with N[5] of the isoalloxazine ring. AtCRY1 has Asp at this position and aCRY and ClCRY4 have Asn (Figs. 2a and S1), whereas dCRY has

Cys416. Consistent with previous studies[37], the C416N and C416D variants of dCRY produced some NSQ, but did so with low yield. As the proton on N[5] ultimately derives from the solvent, we sought to increase solvent accessibility of the flavin pocket. The dCRY L405 position provides the closest solvent barrier to the flavin, but in aCRY and ClCRY4 this residue is a hydrophilic Glu residue. The introduction of L405E into the C416N background greatly increased the NSQ yield (Fig. 2b). Both WT dCRY and L405E/C416N show characteristic absorption bands of oxidized FAD in the dark. Interestingly, the characteristic FAD[ox] absorption peaks at ~450 nm for the L405E/C416N variant appear to be slightly blue shifted (~2 nm) when compared to the WT protein. This spectral shift may result from changes in hydrogen bonding to the N[5] nitrogen on the FAD in the L405E/C416N variant. Upon blue light excitation (5 s, 440 nm, 30 mW), WT converts to the ASQ, with an absorption peak at 367 nm and a shoulder at 401 nm, while the L405E/C416N variant converts to the NSQ as shown by a broad feature

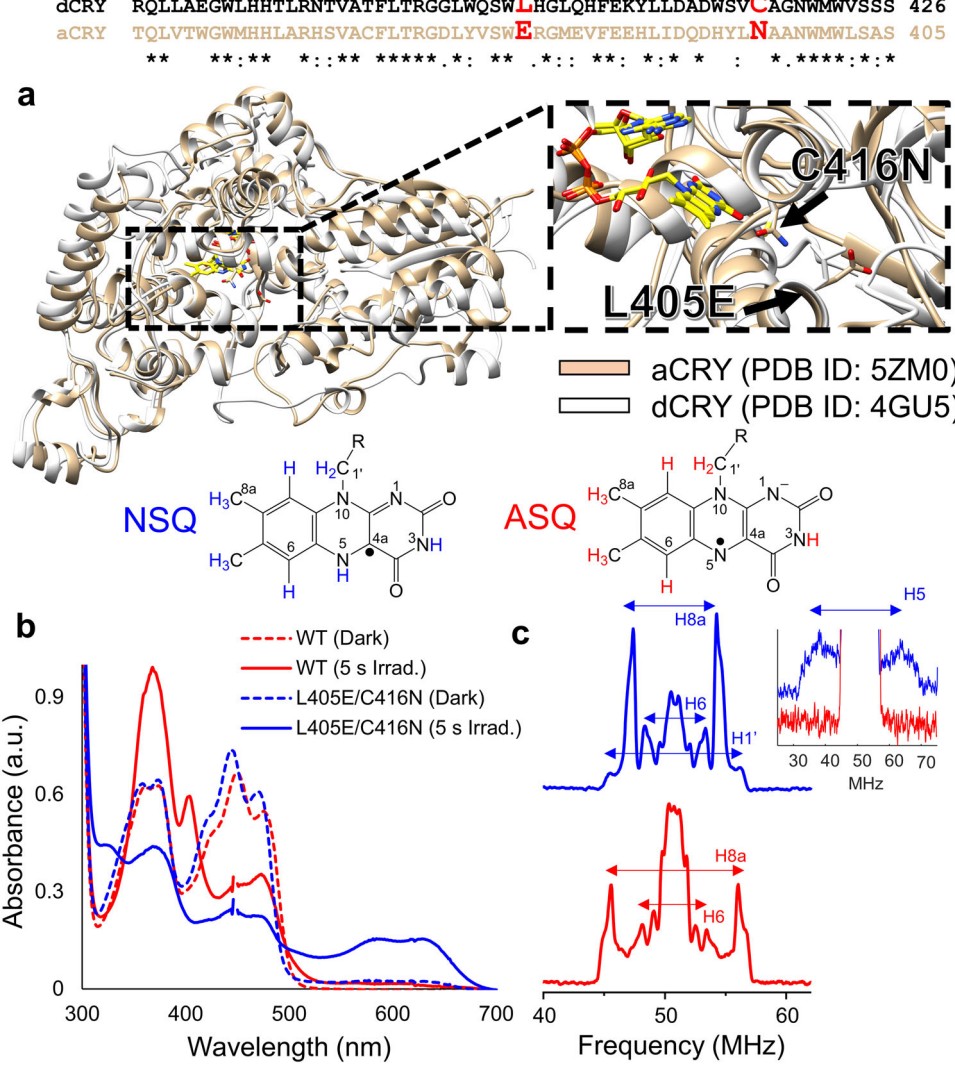

**Fig. 2 An NSQ-forming dCRY variant. a** Structural and sequence alignment of dCRY (gray) and aCRY (tan), key residues that affect the nature of the radical formed are highlighted in red. UV–vis spectroscopy (**b**) and Q-band ENDOR spectra (**c**) of dCRY WT (red) and dCRY L405E/C416N (blue) variant. Frequencies ±1.5 MHz around the central Larmor frequency correspond to solvent protons as well as protons on the isoalloxazine ring (H[3], H[7], and H[9]). The dCRY L405E/C416N variant has hyperfine coupling values at 5.0, 6.8, 10.8, and 25.3 MHz corresponding to the H[6], H[8α], H[1'], and H[5] protons, respectively[39]. The broad hyperfine coupling observed at 25.3 MHz for H[5] (inset) is characteristic of an NSQ radical. In comparison, the H8α and H6 protons of WT dCRY have [1]H frequencies at 10.5 and 5.3 MHz, respectively. The larger hyperfine value of 10.5 MHz for H[8α] in WT dCRY compared to 6.8 MHz in dCRY L405E/C416N is typical of an ASQ[40].

with two maxima at 580 and 632 nm (Fig. 2b). NSQ formation is rapid and complete at pH 6.5–8.0, but at pH 9.0, slowed protonation produces some ASQ intermediate (Fig. S2a).

X-band cw-ESR and Q-band electron-nuclear double resonance (ENDOR) spectroscopy were used to further confirm NSQ formation by dCRY L405E/C416N. X-band cw-ESR spectra of both variants reveal a broader linewidth for L405E/C416N (20.3 G) as compared to WT dCRY (14.7G) (Fig. S2b) which agrees with linewidth values previously reported for ASQ and NSQ radicals[38]. In addition, ENDOR spectroscopy produced hyperfine coupling values for FAD-associated protons (Table 1, flavin numbering scheme in Fig. 2) that are definitive for an NSQ radical[39], which has a different spin distribution over the isoalloxazine ring compared to the ASQ-producing WT dCRY[40] (Fig. 2c). The hyperfine frequencies observed for dCRY WT agree well with previously reported values for the ASQ[14,40]. Additionally, the frequencies observed for dCRY L405E/C416N variant agree well with those for other NSQ forming cryptochromes such as *Synechocystis* species cryptochrome-DASH and *A. thaliana* cryptochrome 1[14,40,41].

**Table 1 Q-band ENDOR hyperfine values and their assignments for dCRY WT and the L405E/C416N (EN) variant.**

| CryWT | Cry EN | Assignment |
|-------|--------|------------|
| n.a. | 25.3 | 1H ($N^5$) |
| n.d. | 10.8 | 1H (1′a) |
| 10.5 | 6.8 | 1H (8) |
| 5.3 | 5.0 | 1H (6) |

For EN, the peak at 25.3 MHz corresponding to the proton on the $N^5$ nitrogen is weak due to a variety of factors such as large anisotropy of the hyperfine tensor as well as the relatively weak power of the r.f. the amplifier in that region. *n.a.* not applicable, *n.d.* not determined.

**C-terminal spin labeling of proteins with sortase A.** ESR spectroscopy is well suited to study biomolecular motion over a wide range of timescales ($10^{-6}$–$10^{-12}$ s$^{-1}$)[42,43]. Nitroxide radicals are commonly used for the analysis of protein motion owing to their spectral response being sensitive to both the fast and slow motional regimes seen in biomolecules. To carry out these measurements, the nitroxide radical (or spin-label) must be attached with high specificity to a given site on a protein of interest. Common spin-labeling approaches include the use of thiol-reactive probes[44], spin-labeled small ligands[45], and the use of radical containing unnatural amino acids[46,47]. For eukaryotic proteins, the presence of multiple exposed cysteine residues often requires the use of cysteine null variants, which may affect protein function and stability. Previously, we attempted to label dCRY with conformationally sensitive probes by targeting reactive Cys and Lys residues; however, lack of labeling specificity and the tendency of the protein to deactivate upon modification or suffer from loss of native cysteine residues largely resulted in failure. In addition, split intein-based approaches succumbed to complications caused by inactive fusion proteins. We then turned to the transpeptidase activity of *S. aureus* sortase A[48–50]. Sortase A catalyzes the ligation of a poly-glycine peptide (GGGG) to an LPXTG motif, where X is any amino acid. The covalent attachment of probe moieties to this poly-glycine peptide (GGGG-Probe) allows its direct C-terminal ligation to an LPXTG motif contained within the protein of interest (POI). The result is a sequence of POI-LPXTGGGG-Probe.

We linked the nitroxide radical *S*-(1-oxyl-2,2,5,5-tetramethyl-2,5-dihydro-1H-pyrrol-3-yl)methyl methanesulfonothioate (MTSL) to a peptide containing a cysteine residue preceded by four glycine residues (GGGG-C-SL, SORTC-SL) (Figs. 3a and S3). Continuous-wave (cw) ESR spectroscopy (Fig. 3b, top spectra) of this glycine probe (SORTC-SL) yields a spectrum in the fast motion regime (0.1–1 ns) with narrow linewidths, as expected for a small (534 Da) peptide rapidly tumbling in solution. As a test

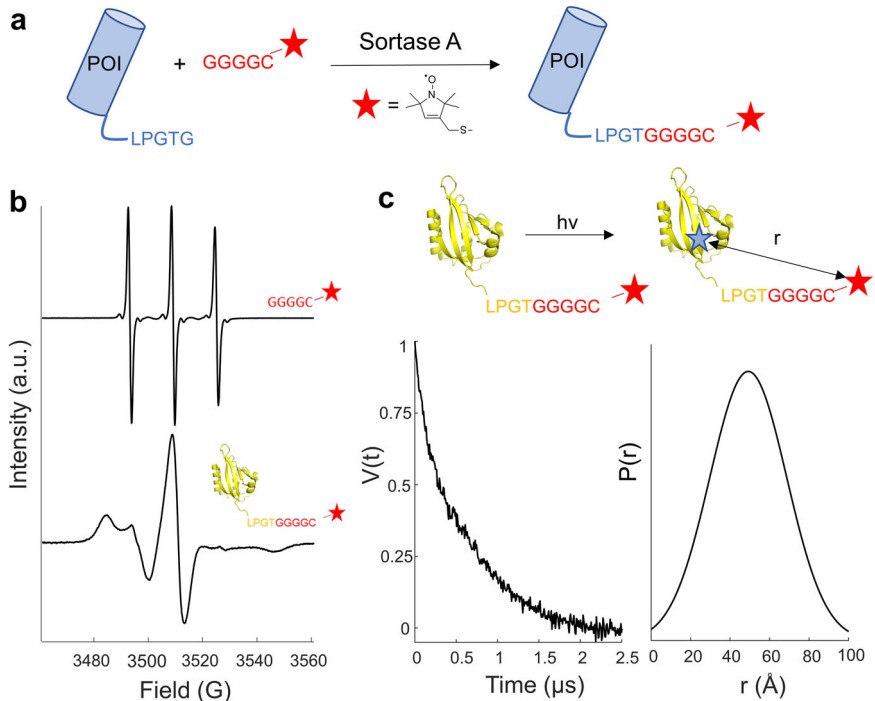

**Fig. 3 Site-specific C-terminal protein labeling using sortase A. a** Sortase A catalyzes the ligation of a polyglycine probe (GGGGC) to an LPGTG recognition motif. For analysis using ESR spectroscopy, the cysteine of the glycine probe was conjugated to a nitroxide radical. **b** cw-ESR spectra of (top) labeled GGGGC peptide, (bottom) C terminally labeled *A. thaliana* iLOV. **c** Photoreduction of iLOV allows distance measurements between the flavin NSQ and sortylated nitroxide radical. DEER measurements (time domain, left) for C-terminally labeled iLOV indicated a broad distance distribution (right).

case, sortase A was used to attach SORTC-SL to the engineered flavoprotein iLOV[51] that contains an FMN cofactor (Fig. 3b). The engineered iLOV protein chosen is relatively small (~12 kDa) and does not dimerize upon light irradiation[52]. The cw-ESR spectra of iLOV-SORTC-SL are broadened when compared to that of the rapidly tumbling SORTC-SL. The broadening is consistent with the slow rotational motion of the nitroxide probe in the range of 1–10 μs, as expected for attachment to a larger protein.

**PDS of flavin-nitroxide radical pairs in proteins**. PDS was then applied in the form of 4-pulse double electron-electron resonance (4P-DEER) to measure distances between the spin centers on the labeled proteins. The method is based on modulating the spin–echo amplitude of the first spin (probe) by flipping the second spin (pump). Successful execution of 4P-DEER between two different radical species such as flavin cofactors and nitroxides depends on (a) the experiment temperature and (b) the choice of radical on which to detect the modulation of spin-echo amplitude. 4P-DEER on flavins are typically carried out at ~120–150 K[53], whereas nitroxides are typically measured between 40 and 60 K[54]. We determined an optimal temperature of 60 K by evaluating signal-to-noise of the different radical species across a range of cryogenic temperatures (Fig. S4). Above 60 K the nitroxide exhibited rapid relaxation (shorter phase memory time), which made it difficult to measure 4P-DEER with sufficient signal-to-noise. The measurements also revealed that nitroxides have longer phase memory times than flavins at 60 K. Hence, we decided to probe the nitroxides and pump the flavin spins. A probe and pump frequency separation of 84 MHz (~30 G) was determined by carrying out a field swept echo experiment on the flavin-nitroxide pair (Fig. S5).

4P-DEER at 35 GHz of iLOV-SORTC after FMN NSQ generation by blue-light illumination (Fig. 3c) produced the expected echo modulation. Transformation of the time-domain signal into a distance distribution of the spin-separation, $P(r)$, reveals a broad distribution with a mean of 4.4 nm and a width of 1.2 nm (Fig. 3c), which would suggest iLOV has a flexible C terminus. The crystal structure of iLOV (PDB 4EES) predicts an FMNH•-to-C-terminal-SL distance of 5.2 nm, (including the 9-residue linker at 0.35 nm per residue). This separation is within the range of the measured distance distribution.

**ASQ formation in dCRY undocks the CTT**. To monitor CTT positioning relative to the flavin cofactor, we used the site-specific labeling approach discussed above combined with 4P-DEER measurements after blue light irradiation for both WT and the L405E/C416N dCRY variant. Nearly complete labeling of dCRY was observed after overnight incubation with SORTC-SL and sortase A, as evidenced by the comparison of protein and nitroxide concentrations. Liquid chromatography–mass spectrometry (LC/MS–MS) identified the C-terminal peptide sequence QFFWLADLPGTGGGGC after trypsin digestion (Fig. S6). We implemented a quantitative fluorescence-based assay to evaluate the light-induced degradation of TIM by sort-tagged dCRY in insect cells. The nitroxide spin-label is unstable in vivo, and thus, we tested a SORTC tag that had the terminal Cys residue replaced by a His to mimic the size of the nitroxide moiety (LPGTGGGGH). Sort-tagged dCRY effectively degrades TIM in the light with similar activity as WT (Fig. S7). Because TIM degradation by dCRY depends on dCRY expression level, which differs among the variants, to aid comparisons we also measured light-induced binding of dCRY to TIM. As expected, WT dCRY binds TIM preferentially in the light, whereas a dCRY-ΔCTT variant binds TIM equally in light and dark. Sort-tagged dCRY also binds TIM considerably better in the light, but the difference

is less than with WT. The additional residues of the SORTC-SL tag may interfere with TIM binding to some extent, but TIM degradation is not greatly affected (Fig. S7).

Blue light induced formation of the ASQ or NSQ states in dCRY(WT)-SORTC-SL and dCRY(L405E/C416N)-SORTC-SL, respectively. After illumination, samples were immediately flash-frozen in liquid nitrogen and transferred to the Q-band pulse spectrometer for 4P-DEER measurements at 60 K. The time-domain 4P-DEER measurements on dCRY variants are immediately revealing in that the WT protein has a slow decay whereas the NSQ forming L405E/C416N variant has a rapid decay with sharp oscillations (Fig. 4a). In 4P-DEER measurements, faster signal decays correspond to shorter distances while oscillations indicate a narrow distance distribution. Therefore, the CTT is closer to the flavin and more rigid in the NSQ-forming L405E/C416N variant compared to the WT. The corresponding distance distributions between the flavin and the nitroxide obtained using DD analysis[55], demonstrate that indeed the ASQ-forming dCRY has a much longer and broader distance distribution of 39.5 ± 6.7 Å when compared to that of the NSQ-forming L405E/C416N variant at 29.3 ± 2.0 Å (Fig. 4b). Furthermore, SVD-based analysis[56] of the 4P-DEER data suggests that the extended state observed for the dCRY WT protein might correspond to rapidly exchanging conformational sub-states (Fig. S8). However, the DD analysis provides the most simplistic model by fitting the dCRY WT protein as a single broad state. This approach allows fitting of the other dCRY variants to a two-state model—docked (NSQ-forming L405E/C416N variant or dark-like) and undocked (ASQ forming WT or light-activated) and is discussed in detail below. Thus, in dCRY containing a photo-reduced ASQ, the CTT releases from the flavin pocket and extends by 1 nm in comparison to the L405E/C416N NSQ-forming variant. This finding agrees well with MD simulations that demonstrated CTT release by ~1 nm upon photoreduction[13].

Next, we probed if dCRY WT undergoes reversible reduction–oxidation cycles; i.e., after a photoreduction cycle of FAD^ox to the SQ radical followed by reoxidation back to FAD^ox in the dark, does the CTT release in the second cycle of photoreduction? Earlier UV–vis and proteolytic protection studies reveal that the dCRY CTT undergoes reversible undocking in light upon the formation of the ASQ radical and re-docking in the dark upon recovery to the FAD^ox state[15]. Furthermore, CTT released by chemical reduction could be recovered and re-released with light[15]. However, we wondered whether subsequent undocking produced conformations similar to those of the initial cycle. Indeed, 4P-DEER derived distance distributions of dCRY WT subjected to reduction–oxidation–reduction conditions are similar to those of the primary photoreduction (Fig. S9), Thus, dCRY is relatively stable to photoreduction followed by reoxidation and the photocycle does not require flavin cofactor release or irreversible unfolding. We do note that in the absence of reductants (required to preserve the nitroxide moiety), dCRY undergoes some oxidative aggregation during recovery (Fig. S10).

To investigate if the observed distance for the NSQ forming L405E/C416N variant corresponds to a docked CTT, we determined the crystal structure of dCRY-SORTC-SL (Table 2). The protein forms a non-physiological disulfide-linked dimer in the asymmetric unit with the predicted positions of the two C-terminal sort-tags located in close proximity to each other at the dimer interface. Only one C-terminal sort-tag can be discerned in the electron density (Fig. S11), with the nitroxide moiety itself conformationally disordered. The SORTC-SL linker orients away from the CTT and dCRY and is, therefore, is unlikely to interfere with the CTT dynamics. The 31–32 Å distance between FAD^ox and the linker cysteine sulfhydryl group (Fig. 4c) is close to the 29 Å distance found by PDS in the NSQ forming L405E/C416N

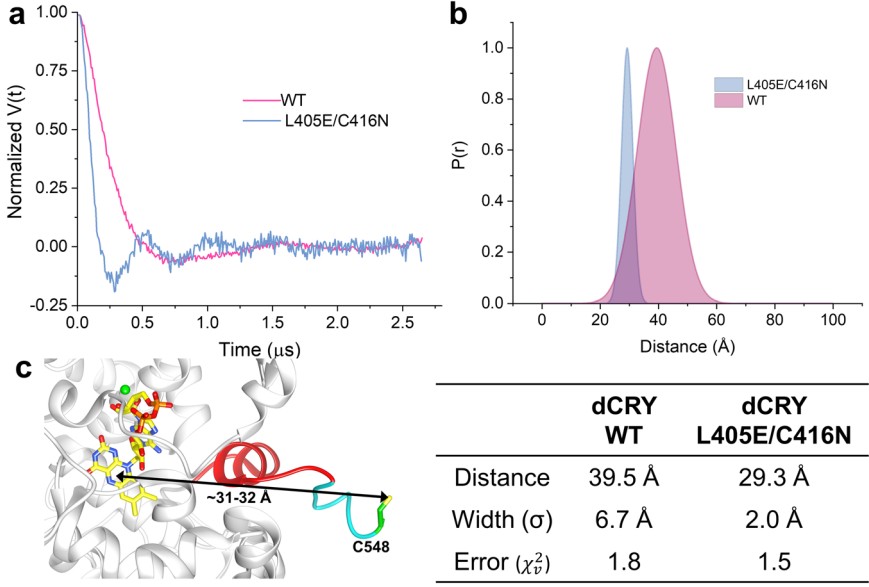

| | dCRY WT | dCRY L405E/C416N |
|---|---|---|
| Distance | 39.5 Å | 29.3 Å |
| Width (σ) | 6.7 Å | 2.0 Å |
| Error ($\chi_v^2$) | 1.8 | 1.5 |

**Fig. 4 Flavin-nitroxide DEER spectroscopy of irradiated dCRY.** 4P-DEER time-domain trace (**a**) and pulse dipolar distance distribution (**b**) obtained between active site flavin and nitroxide radicals on CTT of dCRY variant. The distance distribution is shorter and narrower for the NSQ forming dCRY L405E/C416N (blue) variant when compared to the WT protein (pink) after light exposure. The error values ($\chi_v^2$) for the distance reconstruction are close to 1, indicating a good fit; the error bar on the distance values are below 0.2 Å. **c** The crystal structure of dCRY-WT-SORTC-SL in the dark state shows that the active site FAD and the sulfhydryl group on the GGGGC linker are separated by ~31–32 Å (depending on the FAD atom chosen). Magnesium atoms are shown as green spheres.

variant. The small discrepancy in distance may derive from the unobserved nitroxide moiety residing closer to the flavin than the C-terminal Cys residue or the crystallographic dimer interface influencing the tag position to some extent. From the structure we can also surmise that substitutions at positions 405 and 416 are unlikely to perturb the conformation of the CTT on their own. Both residues are located on the opposite side of the FAD cofactor from the CTT and both E405 and N416 substitutions are predicted to be sterically accommodated by the surrounding residues (Fig. 1b). Overall, the structure indicates that the L405E/C416N variant generally does not release the CTT upon NSQ formation.

**H378 variants affect CTT undocking.** Residue substitutions at H378 alter the interactions of the CTT with the flavin pocket and cause changes to light-induced TIM degradation and dCRY turnover in cell culture. H378N hyperactively degrades TIM, even in dark, whereas H378R and H378K promote light-induced TIM degradation, but also show greatly enhanced dCRY stability in light compared to WT[13]. H378R also degrades less TIM in the light than WT, perhaps indicating that the CTT was especially resistant to activation[13]. Following the same spin–probe strategy as above, we investigated how the H378K/R/N substitutions influenced CTT to release with either a flavin ASQ (light state, undocked) or NSQ (dark state, docked). As expected, H378 substitutions (H378K/N/R) led to ASQ formation while triple substitutions (H378K/N/R, L405E, and C416N) led to NSQ formation as identified by cw-ESR spectral line shapes as well as UV-Vis spectroscopy (Fig. S12). 4P-DEER was performed on all six variants (H378K, H378N, H378R, H378K/L405E/C416N, H378N/L405E/C416N, and H378R/L405E/C416N) after irradiation under identical conditions (Fig. S13). We quantified the relative amounts of the docked and undocked states using DD analysis by fixing the distance parameters from the parent and L405E/C416N proteins (Fig. 4) and varying the relative amounts of the undocked/docked states to fit the 4P-DEER traces. This relatively simple two-state (docked–undocked) model accounts

for the 4P-DEER traces quite well (Fig. S14). Furthermore, to confirm the relative percentages of the two states, we directly fit the time domain traces of the H378 single and triple variants with linear combinations of the time domain traces of the dCRY WT and dCRY L405E/C416N (Figs. S15 and S16). The time-domain fitting and DD analysis are in good agreement (Table S1). As a final check on the DEER analysis, we carried out SVD on all of the dCRY variant data; the additional components retained in the SVD analysis improve fits to the time domain traces (Fig. S17) and the distance distributions (Fig. S18) agree qualitatively with those produced by DD.

PDS of the NSQ forming triple substitutions immediately reveal two distinct distributions that we assign to a docked or undocked CTT, respectively (Fig. 5). Thus, the H378 variants have characteristics of the light state, even in the dark. Substitutions to H378 lead to partial CTT undocking (~40–60% for the L405E/C416N H378N,K,R variants) prior to light excitation (Table 3 and S1). The proportion of undocked CTT in the L405E/C416N background is greatest for H378N (63%), which is consistent with the ability of this variant to reduce TIM levels in the dark. Furthermore, for the single variants (H378K/N/R) we observe some residual docking of the CTT after irradiation, implying that H378 substitutions can partially inhibit CTT release. This effect is most pronounced for H378R. In contrast, H378N shows close to WT levels of CTT activation. The impairment of CTT release may relate to the lack of light-induced dCRY degradation in the H378K and H378R variants, although it should be noted that a considerable fraction of the undocked state is still observed with these variants.

**H378 mutations affect CTT mobility in the dark.** For confirmation that the H378 variants increase C-terminal label mobility in the true dark state (FAD^{ox}) we examined the cw-ESR spectra of the proteins in the absence of a flavin radical. The relative heights of the three spectral peaks that arise from the cw-ESR spectra of nitroxide spin labels (SL) reflect SL mobility. For a freely tumbling SL, these three lines have equal heights, but

**Table 2 Data collection and refinement statistics for dCRY (WT)-SORTC-SL.**

|  | dCRY(WT)-SORTC-SL |
|---|---|
| *Collection statistics* | |
| Wavelength (Å) | 0.97918 |
| Resolution range (Å) | 39.0-2.58 (2.67-2.58)ᵃ |
| Space group | P2₁ |
| *Unit cell* | |
| a, b, c (Å) | 73.0, 122.2, 82.0 |
| α, β, γ (°) | 90, 115.5, 90 |
| *Unique reflections* | |
| Completeness (%) | 96.6 (95.5) |
| Mean $I/\sigma(I)$ | 13.3 (1.5) |
| Wilson $B$-factor (Å²) | 50.4 |
| $R_{merge}$ | 0.114 (0.604) |
| CC1/2 | 0.985 (0.795) |
| *Refinement statistics* | |
| Reflections used in the refinement | 39,481 (3873) |
| Reflections used for $R_{free}$ | 1837 (175) |
| $R_{work}$ | 0.2336 (0.3022) |
| $R_{free}$ | 0.2875 (0.3455) |
| Number of non-hydrogen atoms | 9009 |
| Macromolecules | 8805 |
| Ligands | 109 |
| Solvent | 95 |
| Protein residues | 1086 |
| *RMS deviations* | |
| Bonds (Å) | 0.002 |
| Angles (°) | 0.42 |
| Ramachandran favored (%) | 95.1 |
| Ramachandran allowed (%) | 4.8 |
| Ramachandran outliers (%) | 0.1 |
| Rotamer outliers (%) | 0.4 |
| Clash score | 4.0 |
| Average $B$-factor (Å²) | 57.0 |
| Macromolecules | 57.3 |
| Ligands | 42.0 |
| Solvent | 49.3 |

*Single crystal diffraction. ᵃValues in parentheses are for the highest-resolution shell.

slower rotation resulting from association with a protein will broaden the ESR line shapes and reduce the height of the first (low field) and third (high field) spectral features[43,57].

As expected, all three H378 variants tested (WT vs. L405E/C416N, H378K vs. H378K/L405E/C416N, H378N vs. H378N/L405E/C416N, H378R vs. H378R/L405E/C416N), have comparable dark ESR spectral line shapes in the parent and L405E/C416N background (Fig. S19). These similarities indicate that 1the L405E/C416N substitutions do not greatly affect the nanosecond dynamics of the dCRY CTT in the dark.

However, when compared to either the parent or the L405E/C416N variant, the H378 single variants (H378K/N/R) have faster CTT motion (sharper low field and high field lines and shorter rotational diffusion rate constants, Fig. S19). This data readily agrees with the PDS result that the CTT partially undocks for the H378 variants in the dark state proxy of the L405E/C416N NSQ state. Note that the PDS data measures conformational heterogeneity, whereas the cw-ESR is sensitive to nanosecond motions. We also attempted to study CTT dynamics upon light activation; however, this data is difficult to interpret because the ESR features of the flavin radical overlap with the nitroxide peaks (Figs. S20 and S21). The cw-ESR data combined with the PDS data indicate that changes at H378 largely disrupt the tightly regulated transition from a completely docked to a completely undocked CTT as seen in WT dCRY.

**MDs simulations of the H378K,R variants.** MD simulations of dCRY in the FAD^ox or ASQ states support the assertion that the CTT releases from the PHR following photoreduction[13]. Moreover, simulations of the H378N variant show greater CTT mobility and displacement in the dark state when compared to WT[13]. Because both the biophysical and cellular data indicate that the H378K,R variants behave differently than WT or H378N we also simulated the dynamics of dCRY with these charged residues using the same protocols as before[13]. For H378R, the CTT is stable and docked in the FAD^ox state, but also stable and docked in the ASQ state, qualitatively consistent with the ESR data that shows enhanced CTT docking in the ASQ form. In contrast, the CTT of the H378K variant appears to oscillate among docked, undocked, and more contracted conformations in both the FAD^ox and ASQ states (Fig. S22). This behavior may reflect the time-averaged bimodal populations of docked and undocked conformations observed by PDS. In general, both the simulations and the PDS data indicate that the H378R has a more stable CTT than does H378K, although this trend is more pronounced in the simulations. Inspections of the trajectories indicate that K378 prefers a conformation in which the side chain flips away from the flavin ring and hydrogen bonds to a FAD phosphate group. This movement leaves an opening in the active site that likely causes the CTT destabilization and perhaps more rapid docking and undocking. In one of the H378K trajectories, F534 moves into the cavity vacated by K378, causing a negative jump in the principle component eigenvalue. In contrast, the larger guanidinium group of R378 is unable to flip like K378 and instead shifts down to form favorable electrostatic interactions with the phosphate groups and stack with the W536 indole in what appears to be a very stable configuration (Fig. S23). Thus, interactions of the 378 residues with the CTT depend on the specific properties of the residue—size, hydrogen bonding capability, and charge all influence CTT undocking.

## Discussion

Following conformational changes in photoreceptor proteins is challenging because light-adapted states are often more dynamic and heterogeneous than their more stable dark-state precursors[2,58]. It has been long recognized that the CCEs have important roles in the function that become unmasked by light activations. For the *A. thaliana* CRY1 and CRY2 proteins, expression of the CCEs alone produces constitutive growth phenotypes[59]. Indeed, recent work shows that a specific sequence in the CRY2 CCE mediates interactions with ubiquitin ligase SPA/COP1[60]. The plant CRY CCEs are largely disordered in isolation but structure upon interaction with the PHRs[22]. In the case of plant CRY1, the light-induced formation of a negative charge in the active center involving an aspartate residue that donates a proton to the flavin[61] may expel bound ATP and thereby promote the release of the CCE and the subsequent unveiling of its target sequences[23,62]. However, not all interaction partners of the plant CRYs require the CCE. For example, plant CRY2 binds the basic helix–loop–helix proteins CIB1 in a light-dependent manner, independent of the CCE[63]. The CCEs of mammalian CRYs (mCRYs) have similar functions in molecular recognition, but their conformational states are not linked to light-activation. For example, a region of the mCRY1 CCE coded by exon11 mediates interactions with Clock:Bmal1, and naturally occurring genetic polymorphisms that affect this region alter human sleep behavior[21]. In addition, isozyme specificity of small-molecule inhibitors of mCRYs that target the cofactor binding site depends on the respective CCEs, which again suggests that the CCEs interact with the PHRs[64]. In the case of dCRY, the CTT has been long recognized to be an autoinhibitory element that gates

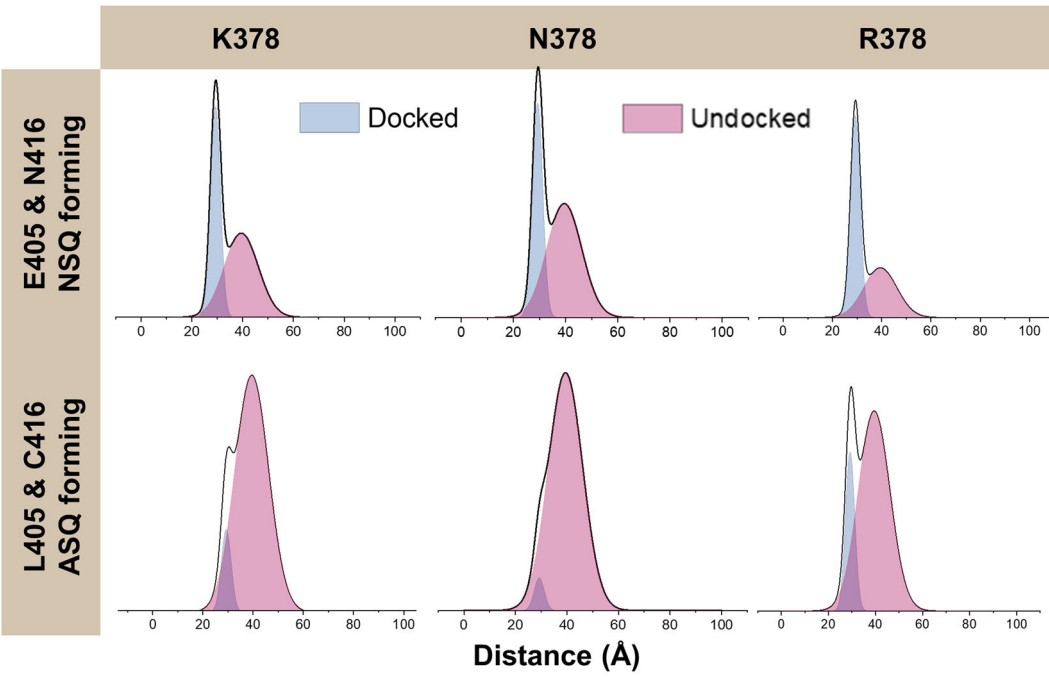

**Fig. 5 H378 substitutions affect CTT conformation in dark and in light.** PDS distance distributions between the flavin and nitroxide radicals for the various H378 dCRY variants. Each spectrum was fit to a combination of docked (L405E/C416N distance distribution) and undocked (WT distance distribution) states. The H378R/K/N variants show a mixture of docked and undocked states even in the dark state proxy of L405E/C416N (top panels).

**Table 3 Percentage of the undocked state observed after light irradiation for H378 variants along with their variance.**

| Variant | Radical | % Undocked | Error ($\chi_v^2$) |
|---|---|---|---|
| H378R | ASQ | 81 ± 0.7 | 2.4 |
| H378K | ASQ | 91 ± 1.6 | 1.5 |
| H378N | ASQ | 95 ± 0.8 | 1.9 |
| H378R/L405E/C416N | NSQ | 47 ± 2.4 | 0.7 |
| H378K/L405E/C416N | NSQ | 57 ± 1.0 | 8.7* |
| H378N/L405E/C416N | NSQ | 63 ± 3.3 | 0.8 |

Residual $X^2$ error estimates are determined in DD and indicate the robustness of fit between the experimental data and the distance reconstruction. A residual $X^2$ error value in the range of 2 or lower is indicative of a good fit. The variance in the population of the undocked state is also given. The H378K/L405E/C416N variant (*) has a relatively high error and is discussed in Fig. S16.

interaction of the PHR with TIM[16–18], as shown by the light-independent binding of dCRYΔCTT (Fig. S7). Photoinduced conformational changes in dCRY have been generally monitored through proteolytic susceptibility and small-angle X-ray scattering[15,26,31] and MD simulations have provided an atomistic description of CTT motion[13,24,31]. However, direct experimental tracking of CTT structure and dynamics in the light state has been lacking.

The L405E/C416N variant produces a change in the flavin photoreduction product from the ASQ to the NSQ and in doing so alters the electronic properties of the flavin cofactor. Remarkably, the CTT has dramatically different properties whether the flavin is negatively charged or neutral. For the ASQ, the CTT label gives longer and broader distributions than when the flavin ring is uncharged. With a neutral flavin ring, the spin–spin distance distribution agrees well with the conformation expected based on the crystal structure of the dCRY dark state. The ASQ

and NSQ also differ by a proton on N[5]; however, protonation is not likely the factor determining CTT release because both the FAD[ox] and the NSQ states have docked CTTs despite differing in N[5] protonation. Although the CTT displaces from its position in the flavin pocket in the dark-state crystal structure state, the signal is not indicative of a fully unfolded CTT, which would produce distances too long and heterogeneous to quantify by 4P-DEER. Thus, even in the light-adapted state, the CTT has a defined structure and remains somewhat associated with the PHR and thereby could provide a structural element for recognizing targets such as TIM, PER, JET, and BRWD3. Consistent with this idea, the sort-tag on the CTT does appear to modestly affect TIM binding in the light (Fig. S7). It is remarkable that the addition of a single negative charge in the flavin active center promotes this substantial conformational change. If the flavin remains neutral, as in the NSQ state, the CTT tightly associates with the protein.

Given its positioning and conservation, H378 likely communicates changes in flavin redox state to the CTT. Previously, we suggested that the H378 protonation state responds to the formation of the ASQ[13]. Asn is a smaller residue than His, but it can supply a hydrogen bond donor in the same position as His N[δ1], which interacts with a flavin ribose hydroxyl group in the crystal structure. However, the H378N variant behaves quite differently than WT in that it degrades TIM in the dark and is hyperactive in the light[13]. The ESR data show that this behavior arises because H378N destabilizes the CTT against the PHR, confirming that CTT movement directly correlates with the ability of dCRY to recognize TIM and facilitate its degradation via JET. Furthermore, the ESR data suggests that this destabilization of the CTT in the H378N variant does not lead to complete undocking of the CTT in the dark but causes the CTT to equilibrate between docked and undocked states. The hydrogen bonding and steric properties of H378 necessary for CTT stability against the PHR cannot be easily reproduced by other residues. Compared to H378N, H378R,K stabilize the CTT against the PHR, even in the light[13]. The effect is most pronounced for H378R, which is also less active in light for TIM degradation[13]. The H378R,K variants

are especially interesting because they block light-dependent dCRY turnover in cell culture[13], although according to PDS results and MD simulations light-induced CTT undocking of H378K does not differ greatly from WT. Thus, light-dependent dCRY turnover may depend on other factors in addition to the CTT displacement. Notably, mammalian CRY1 and CRY2 are targeted by the E3 ligase FBXL3 by the binding of the FBXL3 C-terminus to the exposed (and empty) flavin pocket of CRY1/2[65]. If E3 ligases recognize dCRY similarly, H378 may contribute to this interaction and a change to R or K could then block this process.

CTT undocking requires flavin reduction, but the role of coupled His378 protonation is less clear. MD simulations suggest that H378 protonation undocks the CTT and the pH-dependence of photoreduction rates is lost upon H378 substitution. However, the H378A variant, which cannot undergo protonation, shows little difference in global conformational transitions relative to WT when monitored by time-resolved SAXS over a large range of time scales[31]. The biological activity of the H378A variant is currently unknown as are the details of the structural changes monitored by SAXS. The data presented here indicate that His378 effectively locks the CTT against the PHR in the dark state. Other residues at this position that cannot undergo changes in protonation still allow for the light-dependent triggering of CTT rearrangement, but the docked and undocked states are not converted with 100% efficiency as they are with His. The biological function may require a high-fidelity chemical switch between a fully "docked" and "undocked" state.

After light activation, dCRY is targeted for ubiquitin-mediated degradation[18,20], and thus a given dCRY protein is unlikely to require repeated photocycles to entrain the circadian oscillator. However, not all dCRY functionality depends on interaction with the core clock machinery[66–69]. In particular, dCRY can directly modulate neuron firing frequency in response to light and this property also correlates with flavin photoreduction[68,69]. The role that CTT release plays in neuronal firing frequency is currently unknown, but reversibility of the photocycle may be relevant for these activities. As we have previously shown[15], the conformational changes in dCRY recover upon flavin oxidation and dCRY can undergo repeated cycles of activation. Recent studies indicate that under certain conditions repeated photoactivation of purified dCRY damages the Trp triad, which alters the photoreduction mechanism and may result in flavin loss[24]. Nonetheless, the majority of the protein (>90%) recovers on each photocycle, and some loss of activity is unsurprising given that in vitro dCRY oxidatively aggregates in the absence of reductants. The DEER experiments also indicate that flavin loss is not an obligatory step of the photocycle and dCRY can repeatedly release the CTT during cycles of light activation. It is notable that the ability of dCRY to modulate neuronal firing frequency has a small red-light dependence[68]. As the NSQ would be the only flavin species to absorb at these wavelengths, the red-light effect may involve the conversion of a small pool of inactive NSQ to active fully reduced $FADH^-$, which, like the ASQ, would carry a negative charge.

ESR spectroscopy can provide both dynamical (nanosecond timescale, cw-ESR) and structural information (nanometer range, 4P-DEER), and unlike Förster resonance energy transfer does not require two different, large, and potentially disruptive fluorescent probes. However, 4P-DEER between different radical species, such as flavin-nitroxide, or copper-nitroxide[47,70], is uncommon and more technically challenging than standard nitroxide-nitroxide experiments. 4P-DEER measurements experiments carried out at 60 K by pumping the flavin moiety and probing the nitroxide moiety is an effective protocol for conducting these experiments. Furthermore, rigidly bound flavin probes give narrower distance distributions than those typically observed for spin

probes attached by flexible linkers. Considering that ~0.1–3.5% of proteins are flavoproteins[71], the flavin-nitroxide 4P-DEER method can be used to probe structure and dynamics in a wide range of systems. Moreover, typical cysteine labeling cannot be used for many eukaryotic proteins such as dCRY that contain many reactive cysteine residues. Sortase A peptide ligation is an effective strategy to label the C-terminus of any protein. This approach only requires the addition of an LPGTG peptide sequence to the C-terminus and the labeling reaction involves the pre-modified peptide, thereby preventing undesirable reactions with the protein. Moreover, sortase A labeling is not limited to the C-terminus but can also target the N-terminus or internal loops of the protein[48]. In conclusion, highly specific sortase labeling[48,72], when combined with a radical-forming cofactor, allows a key regulatory element to be referenced to the flavin active center, the source of the chemical signal for affecting conformational change in dCRY.

## Methods

**Expression and purification of sort-tagged dCRY**. Sort-tagged dCRY was cloned into pET28a (Novagen) in frame with an N terminal twin-strep tag and was expressed in CmpX13 cells[73], an engineered strain of *Escherichia coli* that stably expresses a riboflavin transporter. Cells were grown in Terrific Broth (IBI) that was supplemented with 5 μM riboflavin at 37 °C. The cultures were induced with 0.4 mM isopropyl β-D-1-thiogalactopyranoside (IPTG) when the optical density at 600 nm ($OD_{600}$) reached 0.6–0.8 and continued to grow overnight at 17 °C. Cells were harvested and sonicated in lysis buffer containing 50 mM HEPES (pH 8), 150 mM NaCl, 10% glycerol (vol/vol), 0.5 mM Tris(2-carboxyethyl)phosphine (TCEP), 0.5% Triton X-100 (vol/vol), 1 mM $MgCl_2$ with 1 μL of 250 U/μL benzonase, 10 μM aqueous FAD, and 1 mL of 100× protease inhibitor cocktail (PMSF, Leupeptin, Pepstatin-A). The lysate was then centrifuged at 48,000×g for 1 h to remove cell debris, and the supernatant was then added to Strep-Tactin XT resin (IBA). The resin was washed with buffer containing 50 mM HEPES (pH 8), 150 mM NaCl, 10% glycerol (vol/vol), and 1 mM ethylenediaminetetraacetic acid (EDTA). The sort-tagged dCRY was then eluted in wash buffer containing 50 mM biotin, which was a faint yellow. The addition of the sort-tag was made via RF cloning, and all site-directed mutations were made with QuikChange. All DNA sequencing was completed at the Cornell University Biotechnology Center.

**Expression and purification of sortase A and iLOV**. *Staphylococcus aureus* sortase A in pET29b was expressed in *E. coli* BL21 (DE3). Cells were grown in Miller's Luria–Bertani Broth (Difco) at 37 °C and induced at 25 °C with 0.1 mM IPTG and allowed to grow overnight. Cells were harvested and sonicated in lysis buffer containing 50 mM Tris (pH 8), 300 mM NaCl, 2 mM $MgSO_4$ with 1 μL of 250 U/μL benzonase, and 1 mL of 100× protease inhibitor cocktail. The lysis mixture was then spun down 30,000×g for 35 min. The lysate was then added to Nickel–NTA Agarose Beads (Gold Biotechnology) that were pre-equilibrated with 50 mM Tris (pH 8) and 300 mM NaCl. The resin was then washed with buffer containing 50 mM Tris (pH 8), 300 mM NaCl, and 20 mM imidazole. Sortase A was then eluted in buffer containing 50 mM Tris (pH 8), 150 mM NaCl, and 300 mM imidazole. The eluent was then further purified using a preparative Superdex 75 size exclusion column (26/600, GE Healthcare) in 25 mM Tris (pH 8), 150 mM NaCl, and 10% glycerol (vol/vol). pET28-iLOVf was a gift from Brian Smith, University of Glasgow (Addgene plasmid # 63723). iLOV (Q489D) mutation was introduced to iLOV to improve its photoreduction efficiency. iLOV (Q489D) was expressed and purified similar to SortaseA. In short, iLOV was expressed in BL21 (DE3) cells in LB, grown to an O.D. of ~0.7 at 37 °C and induced at 25 °C with 0.1 mM IPTG and allowed to express iLOV overnight. The cells were then harvested, resuspended in lysis buffer (50 mM Tris, 500 mM NaCl, 10% glycerol), lysed by sonication, spun down at 20,000 rpm for 35 min to remove cell debris. The clarified lysate was then loaded onto nickel NTA resin, washed with 10 column volumes of lysis buffer, and then eluted with buffer containing 300 mM imidazole. The elution was collected in fractions, and the fractions that were yellow in color were pooled together, concentrated, and stored.

**Synthesis of spin-labeled SORTC peptide**. The SORTC peptide (GGGGC) was purchased at >98% purity from Biomatik (Cambridge, Ontario, Canada) and MTSL (1-Oxyl-2,2,5,5-tetramethylpyrroline-3-methyl) methanethiosulfonate) was purchased from SantaCruz Biotechnology. The SORTC peptide was dissolved in 50:50 mixture of acetonitrile (AcN) and water containing 50 mM HEPES (pH 8). To this mixture, a slight molar excess of MTSL was added and left to react overnight at room temperature. The reaction was checked by thin-layer chromatography on silica with the mobile phase containing 3:1:1 n-butanol:acetic acid: water. The product SORTC-SL was separated by preparative TLC using the same solvent system, the silica band containing our product was cut out from the plate,

dissolved in a small amount of 50:50 AcN/Water. The product (SORTC-SL) was first spun down to remove any silica/contaminants and the product was checked by LC–MS (mol.wt. ~534). This supernatant was carefully transferred into a round bottom flask and rotovapped. The dry (final) product was finally dissolved in a small amount of buffer (50 mM Tris, 150 mM NaCl, 5% glycerol), aliquoted, and stored at −80 °C.

**Sortylation of dCRY and iLOV.** 5 μM Sort-tagged dCRY was incubated with ~5–6 μM sortase A in a 1:1 molar ratio with excess SORTC-SL in a buffer containing 20 mM Tris (pH 8), 150 mM NaCl, and 5 mM CaCl₂ at 4 °C overnight. This mixture was then further purified using an analytical Superdex 200 size exclusion column (10/300 GL, GE Healthcare) in 50 mM HEPES (pH 8), 150 mM NaCl, and 10% glycerol (vol/vol). The SORTC-SL attachment to the C-terminus of dCRY was confirmed by LC–MS/MS analysis at the Cornell University Biotechnology Center. Sortylation of iLOV was performed in the same manner except a buffer of 50 mM Tris (pH 8), 500 mM NaCl, and 10% (vol/vol) glycerol was used.

**dCRY:TIM interaction studies.** pAc5.1 plasmids of CLIP-CRY variants and TIM-SNAP-HA (HA = hemagglutinin epitope, CLIP, SNAP tags are developed by New England Biolabs) were constructed with the NEB Gibson Assembly method. 10 ml of *D. melanogaster* S2 cells were transfected with 4 μg of total plasmid according to the QIAGEN effectene reagent manual (1.3 μg CRY variants and 2.7 μg TIM). Three days after transfection, cells were split into two culture dishes with 50 μM proteasome inhibitor MG-132 (Santa Cruz Biotechnology, cat. sc-201270) and incubated for 2 h. Then, one culture was exposed to filtered LED light (450–500 nm, ~700 lux) for 1 h while the other one was kept in the dark, then each culture was harvested. Cell pellets were lysed with 1% detergent CA-630 buffer (50 mM Tris, pH 8, 150 mM NaCl, 10% glycerol, and 1% CA-630) with a fresh protease inhibitor cocktail. The lysate was then mixed with 1–5 μM SNAP-tag dye (SNAP-Cell 647-SIR) and CLIP tag dye (CLIP-Cell TMR-Star) according to the NEB manual. For pulldown samples, cell lysate was incubated with anti-HA magnetic beads (Thermo Fisher cat. 88836) overnight at 4 °C. Proteins on beads were incubated with 1–5 μM SNAP-tag dye and CLIP tag dye after being washed with TBST (0.05% Tween-20, pH 7.6). Beads were pulled down with a magnetic rack for wash steps. Proteins were eluted from beads with sodium dodecyl sulfate (SDS) sample buffer and run SDS polyacrylamide gel electrophoresis. Fluorescent bands on protein gels were imaged with ChemiDoc. dCRY, TIM band intensities were quantified by image J-Fiji. As TIM (bait) is saturated by CRY (prey), the affinity difference of the dark and light-exposed samples is calculated as below.

Affinity index = (CRY (prey) on beads)/(TIM (bait) on beads).

Dark-light difference = (affinity index (light exposed sample))/(affinity index (dark sample)).

TIM degradation assays were carried out as above except that no MG-132 added and only lysates were analyzed.

**ESR spectroscopy experiments (continuous wave, ENDOR, and DEER).** Continuous-wave ESR (cwESR) experiments were carried out at room temperature at X-band (~9.4 GHz) with a modulation amplitude of 4 G on a Bruker E500 spectrometer equipped with a super Hi-Q resonator. Light irradiation was carried out by applying a blue laser (448 nm, 30 mW peak power) for 5–7 s directly in the resonator cavity. For the pulse ESR experiments, the samples were further exchanged into deuterated–buffer (50 mM HEPES pH 8.0, 150 mM NaCl) containing 25% glycerol–d8 using a 50 kDa spin column under ambient conditions. X-band by cw-ESR spectra were recorded before and after light irradiation. After irradiation, samples were rapidly plunge frozen into liq. N₂. All pulse-ESR measurements were carried out at Q-band (~35 GHz) on a Bruker E580 spectrometer equipped with a 10 W solid-state amplifier (150 W equivalent TWTA), 150 W RF amplifier, and an arbitrary waveform generator. ENDOR measurements were carried out at 150 K using the Davies sequence (π-t₁-rf pulse-t₂-π/2-t-π-t_echo). The length of the selective microwave pulse was around 60 and 120 ns, respectively for a π/2 and π pulse. For ENDOR, the resonator was critically coupled, and a 150 W Bruker RF amplifier was used to send 20 μs radiofrequency (rf) pulses. DEER measurements were performed at 60 K in an EN 5107D2 Cavity with a cryogen-free insert/temperature controller. DEER was carried out using four pulses (π/2-τ₁-π-τ₁-π_pump-τ₂-π-τ₂-echo) with 16-step phase cycling. The pump (flavin) and probe (nitroxide) pulses were separated by 84 MHz (~30 G).

**DEER spectroscopy data processing.** The DEER data processing was carried out using the SVD method developed at ACERT[56] (https://denoising.cornell.edu/) and with DD version 7B developed at Vanderbilt University[74] (https://lab.vanderbilt.edu/hustedt-lab/dd/). First, we carried out SVD and DD analysis on dCRY(WT)-SORTC-SL and dCRY (L405E/C416N)-SORTC-SL samples to identify the distance parameters. DD was then used to carry out the two-component fits containing the docked (EN variant) and undocked (WT parent) states. In DD, the average distance (⟨R⟩) and the width (σ) for the two components were fixed while varying only the population of two components to find the best fit estimation. We also carried out unrestrained (not fixing ⟨R⟩ and σ) fit in DD for all the variants, most of the fits converged to the ⟨R⟩ > and σ values identified in the WT and EN proteins. The only

exception is the dCRY H378K/L405E/C416N variant (Fig. S16). From DD, in addition to the distance reconstructions and population values, we also obtained the noise corrected the error ($\chi_\nu^2$) values[74], $\chi_\nu^2 = \frac{1}{N-q}\sum_{i=1}^{N}\frac{[V(t_i)-F(t_i)]^2}{s_i^2}$, where $V(t)$ and $F(t)$ are the experimental and the fit data respectively, $N$ is a number of points, $q$ is a number of variables changed and $s_i$ is estimated noise level for the *i*th point. These error values are indicative of fit between the experimental data and the distance reconstruction, values below 2 are good fits. The details of DD fitting and variables have been documented elsewhere[55,74,75].

**UV–vis spectroscopy.** UV–visible spectra were taken of dCRY variants [in 50 mM HEPES (pH 8), 150 mM NaCl, 10% glycerol (vol/vol)] in a quartz cuvette with a pathlength of 2 mm. Spectra were measured by an Agilent 8534 diode-array spectrophotometer with a single reference wavelength set to 800 nm. Samples were illuminated using a blue laser (TECBL-440, 30 mW, 440 nm, World Star Tech) for 5 s.

**Crystallization and structure determination of spin-labeled dCRY.** Crystallization conditions for dCRY-SORTC-SL were variations of the conditions used previously for dCRY[6,7]. Crystals of spin-labeled WT were obtained via hanging drop vapor diffusion by mixing equal volumes of well solution (17% PEG-4000, 150 mM magnesium acetate tetrahydrate, and 100 mM Tris (pH 9) and protein solution (8 mg/mL spin-labeled WT). A dataset diffracting to 2.58 Å was collected at the Advanced Photon Source. Diffraction data were collected on an Eiger16M detector and were indexed and integrated into HKL-2000[76]. Modeling and real-space refinement were carried out in Phenix[77] and Coot[78], respectively, with the structure of the dark-state crystal structure of WT dCRY (PDB:4GU5) as an initial model.

**MDs simulations.** The starting structures of the MD simulations were based on the crystal coordinates of full-length *Drosophila* cryptochrome (PDB ID: 4GU5) similar to the procedure followed by Ganguly et al[13]. dCRY was immersed in an orthorhombic box of rigid TIP3P waters and Na⁺ and Cl⁻ ions were added to produce a neutral physiological salt concentration of 0.15 M. The solvated box was replicated in all three dimensions using periodic boundary conditions and long-range electrostatic interactions were calculated using the particle mesh Ewald method[79] with a cutoff of 12 Å. Bonds involving hydrogen were constrained by the SHAKE algorithm[80]. After equilibrating the system through several stages that held either pressure or volume constant and varied temperature, production trajectories of 25 ns were computed at 298 K in the canonical ensemble (i.e., constant NVT) with a 1 fs time step. A modified Nose–Hoover method in conjunction with Langevin dynamics was employed to maintain constant pressure and temperature during the simulations. All MD simulations were performed using the NAMD program[81] employing the CHARMM22 force field[82] with the CMAP correction[83]. MD simulations were performed with two redox states of the flavin (the neutral state (FAD) and the ASQ state (FAD⁻)). For each scenario, we propagated three independent MD trajectories.

Principle component analysis (PCA) was performed on the motions of the FFW motif residing in the dCRY CTT. Rotational and translational motions from the MD trajectories were removed by first calculating root mean squared fluctuations (RMSFs) of all residues from each independent MD trajectory after aligning each trajectory to the protein backbone of the crystal structure. The average RMSFs of each residue from all the MD trajectories were calculated, and each trajectory was re-aligned based on only those residues that had average RMSFs below a threshold of 1.3 Å. The coordinates of the atoms corresponding to the FFW motif were extracted from these various aligned trajectories and concatenated into a single trajectory on which the PCA was performed.

**Statistics and reproducibility.** All experiments where replications were performed were successful. The physical measurements in the paper were analyzed by state-of-the-art statistical methods, which were used to guide acquisition times and signal averaging. Figure S7 is the summary of three independent experiments.

**Reporting summary.** Further information on research design is available in the Nature Research Reporting Summary linked to this article.

## Data availability

Data supporting the findings of this paper are available from the corresponding author upon reasonable request. The protein X-ray crystallography structure that supports these findings is deposited in the Protein Data Bank (PDB) with accession code 6WTB. Source data for figures is available in Supplementary Data 1 and Supplementary Data 2. The authors declare that all other data supporting the findings of this study are available within the paper and its Supplementary Information files.

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

## Acknowledgements

This work was financially supported by NIH grant R35GM122535 (B.R.C.), NSF grant MCB-1715233 (B.R.C.) and Molecular Biophysics Training grant T32GM008267 (R.D). We thank NE-CAT at the Advanced Photon Source for access to data collection facilities. NE-CAT is supported by NIH/NIGMS awards P30 GM124165 and S10 RR029205. ESR measurements were carried out at ACERT which is supported by NIH/NIGMS awards P41 GM103521 and 1S1 0OD021543. Sortase A plasmid was a gift from William DeGrado at UCSF. We thank Timothée Chauviré for assistance with the ENDOR measurements, the Cornell Biotechnology Resource Center (Ruchika Bhawal, Elizabeth Anderson, Qin Fu, and Sheng Zhang) for assistance with the mass spectrometry data collection and processing, and T. K. Chua for assistance with crystal structure determination.

## Author contributions

S.C., C.M.S., R.D., C.L. and B.R.C. designed the research; S.C., C.M.S., R.D., C.L., C.C.D., and A.G. performed the research; S.C., C.M.S., R.D., A.G., C.L., C.C.D., and B.R.C. analyzed the data; S.C., C.M.S., R.D., and B.R.C. wrote the paper with contributions from all authors. S.C. and C.M.S. contributed equally for this work.

## Competing interests

The authors declare no competing interests.
