## [Peer Review File · Communications Biology]

Tuning flavin environment to detect and control light-induced conformational switching in *Drosophila* cryptochrome

Siddarth Chandrasekaran¹, Connor M. Schneps¹, Robert Dunleavy¹, Changfan Lin¹, Cristina C. DeOliveira¹, Abir Ganguly², and Brian R. Crane^{1*}

¹ Department of Chemistry and Chemical Biology, Cornell University, Ithaca, NY, 14853 USA

² Institute for Quantitative Biomedicine, Rutgers University, Piscataway, NJ 08854 USA

Index for Supplemental Information

Contents

Index for Supplemental Information	2
Table S1: The population of the undocked state observed in the time domain fits.	4
Figure S1. Structural alignment of dCRY, aCRY & C1CRY4	4
Figure S2. Spectral Features of Cry L405E/C416N	5
Figure S3. Purification of Sortase peptide	6
Figure S4. Flavin-Nitroxide – Field Swept Echo	7
Figure S5. Flavin-Nitroxide (DEER): Pump and Probe	8
Figure S6. Mass spectroscopy of dCRY sortylated C-terminal peptide	9
Figure S7. Effect of Sort-tag on TIM binding and degradation by dCRY.....	10
Figure S8. Singular Value Decomposition based distance reconstruction.....	12
Figure S9. Reduction-Oxidation-Reduction	13
Figure S10. UV-Vis spectroscopic studies of dCRY (WT) during photoreduction and reoxidation.....	14
Figure S11. Electron Density Map of the CTT linker.....	15
Figure S12. UV-Vis spectra of dCRY variants	16
Figure S13. 4P-DEER time traces for dCRY variants	17
Figure S14. DD fits to the time domain traces	18
Figure S15. Two components fits in the time domain	19
Figure S16. 4P-DEER of dCRY(H378K/L405E/C416N)-SORTC-SL	20
Figure S17. SVD fits to the time domain traces	21
Figure S18. DEER distance distributions obtained using SVD analysis	22
Figure S19. Cw-ESR Spectra of dCRY variants in the Dark.....	23

Figure S20. Cw-ESR of dCRY WT with C-terminal SORTC-SL before/after light irradiation.....	24
Figure S21. Cw-ESR of dCRY variants after light activation.....	25
Figure S22. MD simulations of the CTT dynamics in H378K/R variants.....	26
Figure S23. MD simulations: A structural perspective	27
Supplementary References	28

Table S1: The population of the undocked state observed in the time domain fits.

Table S1: The percentages of the undocked state observed in the time domain fits. The best fit values obtained from DD has been shown in parenthesis for comparison.

	R378	N378	K378
E405/N416	47 (47)	63 (63)	50 (57)
L405/C416	81 (81)	96 (95)	91 (92)

Figure S1. Structural alignment of dCRY, aCRY & CICRY4

Figure S1: Sequence (top) and Structural Alignment (bottom) of dCRY, CICRY4 and aCRY. The key residues tuning the nature of the flavin radical are highlighted, both CICRY4 and aCRY form NSQ radicals while dCRY forms an ASQ radical.

Figure S2. Spectral Features of Cry L405E/C416N

Figure S2: Spectral features of Cry L405E/C416N. (a) pH dependence of NSQ formation in L405E/C416N. L405E/C416N readily forms the NSQ at pH 6.5-8.0 upon irradiation. However, at pH 9.0 the ASQ forms first and slowly converts into the NSQ, even without constant irradiation. Note that consistent results were difficult to obtain at pH 9.0 owing to aggregation and low protein solubility.

(b) cw-ESR spectra of dCRY WT & L405E/C416N variants after blue light irradiation. Both these protein samples do not contain the CTT nitroxide probe. The L405E/C416N (linewidth = 20.3 G, **blue**) variant has a broader cw-ESR spectrum when compared to the WT (linewidth = 14.7 G, **red**) protein, indicating that the L405E/C416N (EN) variant forms the NSQ radical while the WT protein forms the ASQ radical.

Figure S3. Purification of Sortase peptide

Figure S3: Left - TLC silica plate showing the spin labeling reaction product stained with iodine. Attachment of the SL moiety makes the peptide more non-polar when compared to the unreacted product. The attachment of SL to the cysteine residue has nearly 100% efficiency. Both lanes 2 & 3 are after incubation at room temperature overnight. Right - An LC-MS trace of the final product, showing that the final product contains GGGGC-SL (m/z ~534 Da, SORTC-SL) as the final product.

Figure S4. Flavin-Nitroxide – Field Swept Echo

Figure S4: (a) Field Swept Echo (FSE) of characteristic flavin radicals (ASQ & NSQ) measured at 150K along with nitroxide radicals measured at 60 K. FSE experiments measure the intensity of an echo sequence as the magnetic field is changed, therefore the FSE is similar to an integrated cw-ESR spectra with some differences owing to spin relaxation. The nitroxide radical spectra has its characteristic three peaks (**red**) corresponding to the nitrogen hyperfine coupling while the flavin spectra show a single broad peak. The NSQ (**yellow**) as expected is slightly broader than the ASQ (**blue**) spectra. (b) The FSE of a system containing both ASQ (flavin) & nitroxide radicals measured at 40, 60, 100, 150 K. At 40 K, we can observe the characteristic nitroxide peaks (three peaks) overlapped with a small amount of flavin. At 150 K, we observe only the flavin peak and no nitroxide peaks. Measurements were carried out at 60 K where both radicals have comparable intensities.

Figure S5. Flavin-Nitroxide (DEER): Pump and Probe

Figure S5: Q-band Field swept echo of dCRY (L405E/C416N) after light irradiation containing the C-terminal SORTC-SL motif measured at 60K. There is substantial spectral overlap between the nitroxide and flavin radicals. However, comparing the spectral features of nitroxide and flavin radicals, we chose to pump the flavin radical and probe the nitroxide radical. The pump and probe frequencies are separated by 84 MHz.

Figure S6. Mass spectrometry of dCRY sortylated C-terminal peptide

Figure S6: nanoLC-MS/MS was used to identify the C-terminal peptide – QFFWLADLPGTGGGGC after trypsin digestion of dCRY-SORTC-SL. The mass spectra of the peptide fragment containing the SORTC linker is shown along with the characteristic fragmentation pattern for the peptide.

Figure S7. Effect of Sort-tag on TIM binding and degradation by dCRY

Figure S7: Light-induced dCRY binding to Timeless (TIM) and TIM degradation. (a) Representative gels showing that light (450 to 500 nm, ~700 lux for 1 hour) enhances binding of dCRY to TIM when both proteins are recombinantly expressed in S2 insect cells. TIM is detected by dye addition to a C-terminal SNAP-tag; CRY is detected by dye addition to an N-terminal CLIP-tag (b) Quantification of the dark/light difference of TIM binding by CRY variants (p-values for WT: 0.045, CRY-sortag: 0.041, CRY- Δ CTT is 0.129) ; the results are an average of three independent biological replicates with the individual data points indicated as white circles and the standard deviation shown by the error bars, asterisks indicate $P < 0.05$ for a one-sample two-tailed T-test that the average values are different from 1.0. dCRY lacking the CTT (CRY- Δ CTT) binds TIM equally in the dark and the light, whereas dCRY containing the C-terminal Sort-tag-SL mimic (LPGTGGGGH) shows binding enhancement in the light but less so than does the dCRY WT protein. (c) Representative gels of reduced TIM levels in light (450 to 500 nm, ~700 lux for 1 hour). dCRY itself also undergoes light-induced self degradation. (d) Quantification of three replicates as in (b). TIM levels are significantly reduced in the light when compared to the dark for dCRY WT and Sort-tag, whereas there is nearly no dark-light difference for CRY- Δ CTT (p-values for WT: 0.012, CRY-sortag: 0.045, CRY- Δ CTT is 0.478). Asterisks indicate $P < 0.05$ for a one-sample two-tailed T-test that the average values are different from 1.0. There is no significant difference between dCRY WT and Sort-tag. (e) Relative dCRY expression levels in the dark; note that TIM degradation in c,d depends on overall dCRY levels as well as light. Dark expression levels among different replicates were standardized to exposure level and gel differences by comparing band intensities of standard markers that exhibit some dye binding.

Figure S8. Singular Value Decomposition based distance reconstruction

Figure S8: Singular value decomposition (SVD) based reconstruction of dCRY WT and L405E/C416N flavin-to-CTT probability distribution. (A) and (B) show the PDS time domain fits obtained using the SVD method for the WT and L405E/C416N variant respectively. (C) The distance distribution for the dCRY WT and L405E/C416N variant. The distance distributions obtained using the SVD method agree well with the distributions obtained by the DD method (**Figure 4**). Interestingly, the broad dCRY L405E/C416N distance distribution observed with DD splits into three peaks centered at ~ 3.5 nm, ~ 4 nm and 5 nm. These multiple distances may indicate that the undocked state contains several dominant conformers; however, the features of the broad distribution are dependent on the SVD cutoff threshold; hence the distinct nature of these conformations is not considered substantial.

Figure S9. Reduction-Oxidation-Reduction

Figure S9: dCRY can undergo cycles of CTT release. (a) 4P-DEER Distance distribution of dCRY WT after photoreduction and after undergoing a photoreduction-oxidation-photoreduction cycle. (b) and (c) are the 4P-DEER time traces along with their fits. For the photoreduction measurement, the dCRY WT sample was reduced with blue light for 10 sec and the 4P-DEER was measured at 60 K. This sample was then thawed and then allowed to reoxidize in the dark for 5 h (followed by the disappearance of the ASQ signal). The sample was then re-irradiated with blue light for 10 sec, flash frozen and the 4P-DEER was again measured at 60 K to produce the reduction-oxidation-reduction (ROR) trace. The DEER distance distribution for the ROR sample shows that most of the flavin cofactor remains bound in the pocket and is not released upon recovery or photoactivation. In the ROR sample the flavin-NO radicals have a mean distribution of ~ 42 Å similar to the ~ 39.5 Å seen for the initial photoactivation, The broader distance is unsurprising given that the ROR sample has lower nitroxide (NO^*) radical signal (~ 30 -50% decrease) of the primary sample, owing to the fact that reactive oxygen species produced during reoxidation quench the nitroxide radical to some extent. Furthermore, the freeze thaw cycle causes some protein degradation and aggregation. Both NO^* radical reduction and partial protein aggregation can broaden DEER distance

distributions. Some loss of sample integrity due to the freeze thaw cycle is also indicated by the the ~30% reduction in modulation depth (b, c).

Figure S10. UV-Vis spectroscopic studies of dCRY (WT) during photoreduction and reoxidation.

Figure S10: UV-Vis spectroscopic studies of dCRY (WT) during photoreduction and reoxidation. In the dark, the characteristic FAD^{ox} peaks at ~450nm are observed, which upon photoreduction forms the ASQ radical (peaks at ~367 and 403 nm), this ASQ radical slowly recovers in ~2h in the dark back to the FAD^{ox} , which can then be further photoreduced by blue light. In the ~2h recovery and 5s re-irrad samples, the absorbance values at ~310-320 nm are substantially increased, this increase can be attributed to protein aggregation (increased light scattering at short wavelength by larger aggregated protein particles). Although aggregation can be mitigated by reductants such as DTT/TCEP, these agents also react with the nitroxide spin probes.

Figure S11. Electron Density Map of the CTT linker

Figure S11: The 2.58 Å resolution mFo-DFc composite omit map (blue mesh) of the CTT linker region (LPGTGGGGC) obtained from the X-ray diffraction pattern. The electron density map is contoured at 1.4 σ with a 2.5 Å radius. Electron density (PDB ID: 6WTB) for the linker is clearly present in only one subunit of the structure, although even for this subunit it indicates a less than fully ordered peptide.

Figure S12. UV-Vis spectra of dCRY variants

Figure S12: The UV-Vis spectra of dCRY variants in the dark and after 2s irradiation with 448 nm blue laser. The top row contains the triple variants (H378K/R/N, L405E, C416N) while the bottom row contains the single variants (H378K/R/N). The triple variants all form NSQ radicals (broad feature ~550-650nm) while the single variant all form ASQ radicals (peak ~360 nm).

Figure S13. 4P-DEER time traces for dCRY variants

Figure S13: Q-band 4P-DEER time-domain plots for the dCRY- WT & variants measured at 60 K after light irradiation. All the traces have been background subtracted.

Figure S14. DD fits to the time domain traces

Figure S14: Time domain traces of all dCRY variants and their corresponding fits carried out based on the distance distribution models determined by the DD program. The WT and L405E/C416N were fit with one component whereas the spectra from the other variants were fit using two components. The process for the fitting has been detailed in the methods section and the corresponding distance distributions are shown in **Figures 4 & 5**.

Figure S15. Two components fits in the time domain

Figure S15: The H378 variants were fit in the time domain by a linear combination of the dCRY WT and dCRY L405E/C416N time domain traces. The best fit was calculated by minimizing the difference between the data and fit traces shown. The fitted linear combinations agree well the state weighting obtained in DD (**Table S1**). The H378K/L405E/C416N variant does not fit as well to the experimental data as the other variants, but also has a high error value in the restrained DD fits that is discussed in **Figure S16**.

Figure S16. 4P-DEER of dCRY(H378K/L405E/C416N)-SORTC-SL

	Undocked	Docked
$\langle R \rangle$	39.5	29.3
σ_R	6.7	2.00
%	57	43
χ^2_{ν}	8.6725	

	Undocked	Docked
$\langle R \rangle$	37.20	29.3
σ_R	3.1	2.00
%	64	36
χ^2_{ν}	2.2647	

Figure S16: 4P-DEER distance reconstructions of dCRY H378K/L405E/C416N variant after light irradiation. In the restrained fit the distance parameters such as $\langle R \rangle$ and σ_R were fixed and only the percent of undocked state is allowed to vary whereas in the unrestrained fit all of the parameters are varied. The conformation populations are not very different in the two cases – 57% (restrained) vs 64% (unrestrained). The distance parameters for the docked state are identical in both the fits. The major difference in the two fits is the width (σ_R) of the distance distributions; in the unrestrained fit the undocked state converges to a narrower width of 3.1 Å as opposed to a width of 6.7 Å observed in the dCRY WT parent. As the other variants converge well to the dCRY WT parent distance distributions on performing unrestrained fits, we think that the restrained fit using the dCRY WT parent is a better approach to the distance constructions. It is possible that the narrower distribution observed in the unrestrained fit is real and indicates that the H378K mutation causes the CTT lock in an unnatural conformation.

Figure S17. SVD fits to the time domain traces

Figure S17: The time-domain fits performed by SVD compared to experimental spectra.

Figure S18. DEER distance distributions obtained using SVD analysis

Figure S18: Distance distributions obtained using SVD analysis for each of the dCRY variants. These distance distributions follow the general trend observed in the DD analysis (**Figure 5**), however quantification of the docked and undocked states is less straightforward owing to the development of spurious peaks which depend on the threshold for the singular values retained in the reconstruction².

Figure S19. Cw-ESR Spectra of dCRY variants in the Dark

Figure S19: (a) Cw-ESR Spectra of indicated dCRY variants acquired in the dark. Rotational time constants (τ_c in ps/revolution) for each spectra were calculated according to Kovarski et al¹. (b) Average rotational time constants for each residue at position 378 indicate that mutations to K, N, and R have increased mobility (decreased values of τ_c) by around a factor of 2 compared to H378.

Figure S20. Cw-ESR of dCRY WT with C-terminal SORTC-SL before/after light irradiation

Figure S20: X-band cw-ESR of dCRY(WT)-SORTC-SL in the dark (D) and after light irradiation (L). As seen clearly, the flavin radical spectra overlap substantially with the nitroxide radical spectra.

Figure S21. Cw-ESR of dCRY variants after light activation

Figure S21: X-band cw-ESR spectra of dCRY parent containing the SORTC-SL peptide after light irradiation measured at room temperature. All these spectra contain two radical species, a nitroxide radical containing three relatively narrow peaks and a broad flavin radical. The ESR spectra of both the radicals overlap with each other. Note: The WT and single variants contain an ASQ flavin radical (narrower spectra), while the double and triple variants (broader spectra) form an NSQ flavin radical.

Figure S22. MD simulations of the CTT dynamics in H378K/R variants

Figure S22: The eigenvalue of the first PC that corresponds to CTT undocking is plotted along the trajectories from the MD simulations of the H378K/R variants. Each subplot contains three independent trajectories of ~70 ns long. Greater fluctuations in the plot indicate that the CTT is more flexible.

Figure S23. MD simulations: A structural perspective

Figure S23: The relative orientations of the R378 (a,c) and the K378 (b,d) residues with respect to the CTT and the reduced FAD in the active site. R378 interacts with the FAD and the indole of W536, whereas the K378 residue moves down to engage the phosphate groups of flavin group leaving a vacancy for the F534 side chain to fill.

Supplementary References

1. Kovarskii, A. L., Wasserman, A. M. & Buchachenko, A. L. The study of rotational and translational diffusion constants for stable nitroxide radicals in liquids and polymers. *J. Magn. Reson.* 7, 225–237 (1972).